# An Integrated Wireless Multi-Sensor System for Monitoring the Water Quality of Aquaculture

**DOI:** 10.3390/s21248179

**Published:** 2021-12-07

**Authors:** Jen-Yung Lin, Huan-Liang Tsai, Wei-Hong Lyu

**Affiliations:** Department of Computer Science and Information Engineering, Da-Yeh University, Changhua 515006, Taiwan; jylin@mail.dyu.edu.tw (J.-Y.L.); R0906007@cloud.dyu.edu.tw (W.-H.L.)

**Keywords:** wireless multi-sensor system, freshwater aquaculture, ThingSpeak IoT, ThingView APP

## Abstract

Water temperature, pH, dissolved oxygen (DO), electrical conductivity (EC), and salinity levels are the critical cultivation factors for freshwater aquaculture. This paper proposes a novel wireless multi-sensor system by integrating the temperature, pH, DO, and EC sensors with an ESP 32 Wi-Fi module for monitoring the water quality of freshwater aquaculture, which acquires the sensing data and salinity information directly derived from the EC level. The information of water temperature, pH, DO, EC, and salinity levels was displayed in the ThingSpeak IoT platform and was visualized in a user-friendly manner by ThingView APP. Firstly, these sensors were integrated with an ESP32 Wi-Fi platform. The observations of sensors and the estimated salinity from the EC level were then transmitted by a Wi-Fi network to an on-site Wi-Fi access point (AP). The acquired information was further transmitted to the ThingSpeak IoT and displayed in the form of a web-based monitoring system which can be directly visualized by online browsing or the ThingView APP. Through the complete processes of pre-calibration, in situ measurement, and post-calibration, the results illustrate that the proposed wireless multi-sensor IoT system has sufficient accuracy, reliable confidence, and a good tolerance for monitoring the water quality of freshwater aquaculture.

## 1. Introduction

Suffering from the COVID-19 pandemic since 2020, both fisheries and aquaculture businesses have been facing a range of operational difficulties, such as labor, financing, logistics, and weak market procurement, all affecting the output reduction in 2020. As vaccine roll-out programs continue all over the world, fish production is expected to have a significant boost and positive growth. As pointed out by the Food and Agriculture Organization of the United Nations (FAO) [1], aquaculture has already demonstrated its crucial role in global food security with the production growing rate of 7.5%/year since 1970. As shown in Figure 1, the global fish consumption is ever-increasing, and fish production from aquaculture was of significant importance between 2000 and 2018. Notably, the fish production from inland aquaculture in 2018 was twice that in 2000. Therefore, both marine and inland aquaculture is significantly considered as an integral component for global food security and economic development. This implies aquaculture requires intensive care in the culture process to enhance yield, including stocking, feeding, and protection from predators and disease. The water quality of seawater and freshwater for aquaculture is essential. Therefore, in situ measurement instruments and water quality monitoring systems are of increasing importance for real-time and cost-effective aquaculture management.

Recently, many researchers have paid attention to water quality monitoring. Zhua et al. [2] demonstrated an online water quality monitoring system with temperature, pH, dissolved oxygen (DO), and electrical conductivity (EC) sensors for intensive fish culture, wherein the EC level was adopted to estimate the salinity level highlighted as “*” in Table 1. Based on the collected data, a DO forecast model with the ability of half hour-ahead was further designed using a convolutional neural network (CNN) stochastic gradient descent (SGD) with a momentum (SGDM) algorithm. Zhang et al. [3] used an Orion 5-Star Portable pH/ORP/DO/EC Multimeter to measure the levels of temperature, pH, DO, and EC for an integrated recirculating aquaculture system (RAS) in a constructed wetland for land-based fish farming. A smart wireless mesh sensor network called AquaMesh, with temperature, pH, DO, and EC sensors, was developed by Odey and Li for aquaculture [4]. Simbeye and Yang demonstrated a LabVIEW-based water quality monitoring and control system for aquaculture based on a wireless sensor network (WSN), which had the reading functions of temperature, pH, DO, and EC and had the ability to control pH level, water, and aerator pumps [5]. Schmidt et al. [6] addressed a low-cost, compact, autonomous water quality monitoring buoy system for coastal aquaculture, which used three temperature sensors, one DO sensor, and one pressure sensor arranged to acquire sensing data and stored in built-in data loggers. Then, the sensing data were manually offloaded to display using SOHO software. Saparudin et al. simply used water temperature sensing to demonstrate a proof-of-concept wireless water quality monitoring system for high density aquaculture [7]. Danh et al. demonstrated an IoT-based water quality monitoring system with the sensing ability of water temperature, pH, DO, and salinity levels for aquaculture [8]. To the best of our knowledge, the published papers are compared and tabulated in Table 1. From the above review, water temperature, pH, DO, EC, and salinity levels are important factors for freshwater aquaculture, and salinity level could be estimated using the EC value using the empirical relationship which is addressed in this work. It should be noted that the work of Zhua et al. [2] did not present the detail about salinity level estimation using the EC value. ZeeBee, Wi-Fi, and coexistent mobile data communication techniques such as GPRS/3G and current 4G/5G could be adopted as a WSN platform. Two types of technologies for the service layer in Internet of Things (IoT) are private intranet [2] and public IoT APIs [4,7,8]. With the ever-increasing developments in WSNs, integrated multiple sensors with commercially available Wi-Fi modules such as ESP-32 featuring cost-effectiveness, compactness, and flexible input/output (I/O) interfaces are commercially available. These motivated us to integrate commercially available temperature/pH/DO/EC sensors and a cost-effective ESP-32 Wi-Fi module to develop a wireless multi-sensor water quality monitoring system for aquaculture in the ThingSpeak IoT API, where the water temperature, pH, DO, EC, and salinity information can be visualized in a user-friendly manner through an online browser or the ThingView APP.

The main contribution of this paper is the integration of the water temperature, pH, dissolved oxygen (DO), and electrical conductivity (EC) sensors with Wi-Fi wireless communication to illustrate and demonstrate the above sensing data and estimated salinity level information by the EC level in the ThingSpeak IoT platform for freshwater aquaculture; this has been illustrated and evaluated with sufficient accuracy and confidence. The remainder of this paper is organized as follows. First, the system design and the related theory are described in Section 2. Section 3 reveals the implementation of the proposed wireless multi-sensor system by integrating temperature, pH, DO, and EC sensors with an ESP-32 Wi-Fi module to provide the above readings and an estimated salinity level for monitoring the water quality of aquaculture. The evaluation results are illustrated in Section 4. Finally, brief conclusions and future works are drawn in Section 5. 

## 2. Wireless Multi-Sensor IoT System Design

As shown in Figure 2, the proposed wireless multi-sensor water quality monitoring system for aquaculture consists of temperature, pH, DO, and EC sensors, an ESP32 Wi-Fi module and Wi-Fi access point (AP), as well as the ThingSpeak IoT API. With the availability of the EC level, the salinity level can be directly estimated and the information on the water temperature, pH, DO, EC, and salinity levels can be simultaneously visualized in the ThingSpeak IoT platform and by the ThingView APP. Accuracy, reliability, easy maintenance, and expandability are essential for the design requirements of the proposed in situ remote aquaculture monitoring system. Therefore, sensors with a high sensitivity should be chosen for data accuracy and confidence. Some parts in the sensor are replaceable, such as the filling solution and electrode probe, lowering maintenance costs. The modularization of the sensors, ESP32 Wi-Fi module, and Wi-Fi AP make the overall system highly reliable and easy to maintain. The proposed system features expandability attributing to both multiple general purpose I/O (GPIO) interfaces in ESP-32 and configurable channels in the ThingSpeak IoT platform.

### 2.1. Sensors Description for Aquaculture

As shown in Figure 3, the temperature, pH, DO, and EC sensors are integrated in the proposed multi-sensor system. Figure 3a shows a waterproof 1-wire DS18B20 temperature sensor which is a DS18B20 digital thermometer encapsulated in a 6 × 50 cm stainless steel tube with its three pins are jacketed in PVC [9]. The DS18B20 digital thermometer consists of four main data components: (1) a 64-bit lasered ROM; (2) a temperature sensor; (3) nonvolatile temperature alarm triggers TH and TL; and (4) a configuration register. In order to perform accurate temperature conversions, information is sent–received over a 1-wire interface “*D*_Q_” with external power input from the other two pin ports: “*V*_DD_” and “GND”. Based on such a pin configuration of the DS18B20 digital thermometer, the external power causes other data traffic on the 1-wire “*D*_Q_” port during the active conversion time and other DS18B20s to be placed on the same 1-wire bus. The DS18B20 digital temperature probe was adopted to measure the water temperature with a wide range from −55 °C to 125 °C and high accuracy of ±0.5 °C @ −10 °C to 85 °C.

pH is a quantitative unit of measure on a scale of 0 to 14 that describes the degree of acidity or alkalinity of a substance. The formal definition of pH is the negative logarithm of the hydrogen ion concentration [H^+^], i.e., pH = 10 log([H^+^]), which is the hydrogen ion activity. As shown in Figure 3b, the pH meter is itself a potentiometer containing both sensing and reference electrodes to read the potential difference between the sensing and reference electrodes. The standard solution of the reference is a liquid with the hydrogen mole concentration of 1 mol/L. To find the effect of temperature on the pH reading, the pH meter with a pH electrode is first calibrated using at least two pH buffers that bracket the pH values of samples. The calibration is programmed with the expected pH values of the pH buffers at different temperatures based on the ISO 10523 standard [10] as shown in Table 2. Furthermore, the relationship between the potential and hydrogen ion activity in the sample, described by the Nernst equation, is given as
(1)$$E_{\mathrm{pH}}=E_{\mathrm{pH}0}-2.3\frac{RT}{nF}log(\left[H^{+}\right])$$
where *E*_pH_ is the total potential (mV) between the sensing and reference electrodes, *E*_pH0_ is the standard potential of the reference electrode in the standard solution, *R* is the gas constant (= 8.314 J K^−1^ mol^−1^), *T* is the temperature (K), *n* is the number of ion carriers, F is the Faraday constant (= 96,485 C mol^−1^), and [*H*^+^] is the concentration of hydrogen ions in the sensing solution. Taking the hydrogen ions into consideration, the readings relationship between the potential of the pH electrode and pH value can be rewritten as
(2)$$E_{\mathrm{pH}}=E_{\mathrm{pH}0}+0.23\gamma_{\mathrm{PH}}\mathrm{pH}$$
where *γ*_pH_ is the temperature-related compensation factor of the pH reading (in mV/pH). For the temperature of **0** °C, 25 °C, 50 °C, and 100 °C, the values of *γ*_pH_ are 54.20, 59.16, 64.12, and 74.04 mV/pH, respectively. An operational amplifier (OP Amp.) circuit is used to linearly amplify the output voltage of the pH probe as shown in Figure 3b. An Analog pH Sensor/Meter Pro Kit V2 for Arduino with industrial an pH electrode and BNC connector [11] was used to acquire the pH value of the water quality for aquaculture. The analog pH sensor is itself a potentiometer containing both sensing and reference electrodes to read the potential difference between the sensing and reference electrodes. The measuring range of the analog pH sensor is between 0 and 14 with an operation temperature in the range of 55 to 125 °C and an accuracy of ±0.1 (@25 °C).

DO is oxygen (O_2_) dissolved in water in proportion to the partial pressure of O_2_ in the atmosphere, which can be expressed as the amount of O_2_ dissolved per unit volume of water (mg/L). DO level is of upmost importance to a great deal of aquatic life, including fish, invertebrates, bacteria, and plants which make use of oxygen in respiration similar to organisms on land. Bottom feeders, crabs, oysters, and worms have minimal amounts of biological oxygen demand (BOD) (1–6 mg/L), while shallow water fish have higher BOD levels (4–15 mg/L) [12]. It is known that the saturation level of O_2_ dissolved in pure water at 25 °C and 1 atm (1013 kPa) is 8.11 mg/L. The types of DO sensor can be categorized as electrochemical (such as galvanic or polarographic DO sensors) and optical sensors. The factors of aquatic temperature, salinity, barometric pressure, and flow could affect the electrochemical DO reading; however, only the temperature effect on the DO level is considered due to the temperature sensor being available. A commercially available galvanic DO sensor (Analog Dissolved Oxygen Sensor) with the measurement range of 0–20 mg/L and an operation temperature range between 0 °C and 50 °C was used to measure the aquatic DO level [13]. As shown in Figure 3c, when a galvanic DO sensor is immersed in water, the anode electrode is oxidized and releases electrons and the cathode electrode passes electrons to have a reduction of dissolved oxygen in water. The oxidation and reduction reactions occur as follows.
(3)$$\mathrm{Anode}\;\mathrm{oxidation}\;\mathrm{reaction}:\;2\mathrm{Zn}\to {2\mathrm{Zn}}^{2+}+4e^{-}$$
(4)$$\mathrm{Cathode}\;\mathrm{reduction}\;\mathrm{reaction}:\;O_{2}+4e^{-}{+2H}_{2}O\to {4\mathrm{OH}}^{-}\;{4\mathrm{OH}}^{-}{+2\mathrm{Zn}}^{2+}\to 2\mathrm{Zn}{\left(\mathrm{OH}\right)}_{2}$$
(5)$$\mathrm{Total}\;\mathrm{reaction}:\;2\mathrm{Zn}+O_{2}+H_{2}O\to 2\mathrm{Zn}{\left(\mathrm{OH}\right)}_{2}$$

The resulting current is proportional to the partial pressure of oxygen, which is calculated by the following equation [14]:(6)$$i_{\mathrm{DO}}=\frac{4F\mu_{\mathrm{MB}}\left(T\right)A_{\mathrm{MB}}p_{O_{2}}}{d_{\mathrm{MB}}}$$
where *μ*_MB_(T), *A*_MB_, and *d*_MB_ are the permeability, area, and thickness of the membrane and $p_{O_{2}}$ is the partial pressure of O_2_.

As shown in Figure 3d, EC is a measure of the ability of water to conduct electrical current due to aquatic dissolved and other organic chemicals carrying electrical charges. The EC of aquatic water can be defined as
(7)$$\sigma =q\overset{N}{\underset{i=1}{\sum }}\left(\mu_{i}^{+}+\mu_{i}^{-}\right)\eta_{i}$$
where q is the electrical charge (=1.602 × 10^−19^ C), $\mu_{i}^{+}$ and $\mu_{i}^{-}$ are, respectively, the mobility of positively and negatively charged ions, and ηi is the ion concentration. The EC of ions in the aquatic water is highly temperature-dependent as in the electron conductivity of metallic conductors; however, the aquatic EC typically raises with increasing temperature. The temperature-related EC can be described as
(8)$$\sigma_{\mathrm{EC}}\left(T\right)=\sigma_{\mathrm{EC}}\left(T_{\mathrm{Ref}}\right)\left[1+\gamma_{\mathrm{EC}}\left(T-T_{\mathrm{Ref}}\right)\right]$$
where σ_EC_(*T*) and σ_EC_(*T*_Ref_) are, respectively, the EC values for the specific and reference temperatures *T* and *T*_Ref_. In addition, *γ*_EC_ is the temperature-related compensation factor which ranges from 3 to 1 and in the most naturally occurring samples of water is about 2%/°C. Having the EC level, the corresponding salinity value can be estimated by the following equation:(9)$$\eta_{\mathrm{Sal}}=640\sigma_{\mathrm{EC}\left(T\right)}$$
where *σ*_EC(T)_ is the EC value in mS/cm [15]. For reading the EC value of freshwater aquaculture, an EC sensor (Analog Electrical Conductivity Sensor/Meter V2 (K = 1)) [16] with a measuring range of 0–20 ms/cm, an accuracy of ± 5% F.S., and an operation temperature range between 0 °C and 40 °C was selected. The main specifications of temperature, pH, DO, and EC sensors for the proposed wireless water quality monitoring system are listed in Table 3. Referring to the water quality indicators for aquaculture [17,18], the warning ranges of the corresponding sensors are also predefined and tabulated in Table 3.

### 2.2. Hardware Description for the Wireless MCU and IoT System

Furthermore, the ESP32 is a successor to the ESP8266, featuring the integration of low-cost, low-power system-on-a-chip (SoC) microcontrollers integrating with Wi-Fi and dual-mode Bluetooth. A commercially available ESP32 module, ESP-WROOM-32, created by Espressif [19], was adopted as the microcontroller platform in charge of receiving the sensing data from the temperature, pH, DO, and EC sensors and forwarding them to the ThingSpeak IoT API through a neighboring Wi-Fi AP. The pin assignments of the ESP-WROOM-32 Wi-Fi module include both the GPIOs and the power supply of 5 V_DC_ and GND for the temperature, pH, DO, and EC sensors as shown in Figure 2. The programmable ESP-WROOM-32 SoC-based microcontroller was designed to feature the functions of both automatic warning and line notify, which can be implemented by the code programming as explained later. The warning ranges of the temperature, pH, DO, and EC sensors are listed in Table 3. For the Wi-Fi AP, its SSID and password are checked in advance and are coded in the software program later. 

The open-source IoT application program interface (API), well-known as “ThingSpeak IoT” [20], conveniently provides an IoT platform to store and retrieve sensing data from temperature, pH, DO, and EC sensors using the Message Queuing Telemetry Transport (MQTT) publish–subscribe network protocol and application layer Hypertext Transfer Protocol (HTTP) over the Internet. The ThingSpeak IoT API provides the sensing data logging and update application as well as the geographic information system (GIS) for the location tracking application. The live data streams in the cloud can be further aggregated, visualized, and analyzed using MATLAB software. The hardware implementation of the proposed wireless water quality monitoring system for aquaculture is illustrated in Figure 2.

## 3. System Implementation

### 3.1. Hardware Implementation for the Wireless Multi-Sensor IoT System

Figure 2 reveals the hardware implementation for the proposed wireless multi-sensor IoT system for monitoring the water quality of freshwater aquaculture, which consists of DS18B20 temperature, pH, DO, and EC sensors, acquired signal converters, an ESP-WROOM-32 Wi-Fi module and Wi-Fi AP, and the ThingSpeak IoT platform. As shown in Figure 2, the commercially available DS18B20 temperature sensor, Analog pH Sensor, Analog Dissolved Oxygen Sensor, Analog Electrical Conductivity Sensor, and the corresponding signal converters are arranged in parallel to connect with an ESP-WROOM-32 Wi-Fi module which simultaneously provides the 5 V_DC_ and signal ground (GND) for the sensors. In addition, the ESP32 Wi-Fi module further computes the corresponding salinity level by the acquired EC level. Firstly, a 4.7 kΩ resistor is added between the digital signal pin *D*_Q_ and the 5 V_DC_ power source to make the temperature reading transfer stable as shown in Figure 3a. The reading value of water temperature was further used to automatically compensate the fluctuation output of the pH/DO/EC electrodes. The built-in automatic temperature compensation (ATC) function of the software cost-effectively facilitates the quick and accurate results of pH/DO/EC readings as well as the salinity estimation for in situ application. The BNC connectors for the pH/DO/EC electrodes are directly connected to the corresponding signal conversion transmitters, respectively. The output values are then read by the ESP-WROOM-32 Wi-Fi module. The pin assignment of the ESP-WROOM-32 Wi-Fi module for the temperature, pH, DO, and EC sensors is tabulated in Table 4. The ESP32 Wi-Fi module forwards the data of water temperature, pH, DO, EC, and salinity level through the on-site Wi-Fi AP to the ThingSpeak IoT platform. The information of the water temperature, pH, DO, EC, and salinity level can be visualized by online browsing or by the ThingView APP.

### 3.2. Software Design for the Wireless Multi-Sensor IoT System

The design of the ESP-WROOM-32 SoC-based microcontroller can be programmed in the same way as Arduino platforms under the open-source Arduino integrated development environment (IDE), which supports the languages C and C++ with specified rules of code structure. After the installation of the CP2102 USB drive and ESP drive files in the Arduino IDE environment, the software for the proposed wireless multi-sensor water quality monitoring system was developed in the same coding configuration in the same platform. After including the associate libraries, accessing the configuration of the ThingSpeak IoT platform, and constant/variable declaration, the Wi-Fi AP access and the start of the pH/DO/EC sensors are initialized. Then, the temperature, pH, DO, and EC sensors begin to visualize the water temperature and pH/DO/EC levels, the corresponding ATC calibrations for the pH/DO/EC readings are conducted, and the salinity level is estimated by the calibrated EC level. The values of water temperature, pH, DO, EC, and salinity levels are sequentially arranged and sent to the predefined channel in the ThingSpeak IoT platform. The visualization of water quality is conducted every 15 min automatically without external interruption. The observations can be visualized in the ThingSpeak IoT Platform as shown in Figure 4. The observation data are analyzed based on the MATLAB environment. The same information can be visualized by the ThingView APP as shown in Figure 5. The flowchart of data acquisition for the proposed wireless multi-sensor module is depicted in Figure 6. It should be noted that the ThingSpeak IoT API provides the line notify function through ThingHTTP Apps online with user-friendly configuration interfaces. In addition, the code for the Line Notify function was directly programmed and downloaded into the ESP-WROOM-32 Wi-Fi module in the proposed water quality monitoring system for freshwater aquaculture.

## 4. Results and Discussion

In order to double-check the accuracy, confidence, and reliability of the proposed water quality monitoring system for aquaculture, the temperature, pH, DO, and EC sensors were calibrated in advance, then the wireless multi-sensor system conducted an in situ evaluation for 20 consecutive days, and finally the sensors were post-calibrated to check the possible drift in instrument accuracy. The associated processes are described as follows.

### 4.1. Pre-Calibration Process

Before integrating the temperature, pH, DO, and EC with an ESP-WROOM-32 Wi-Fi module, the sensors can be calibrated in advance. A HEL-711 3-wire Resistance Temperature Detector (RTD) with an accuracy of ±0.3 °C (@ −40 °C to 100 °C) [21] was adopted to calibrate the waterproof DS18B20 temperature sensor. Both the DS18B20 temperature sensor and HEL-711 RTD were put in a cup of distilled water for 24 h with a 10 min sampling rate. In order to calibrate the pH sensor prior to in situ measurement, the Analog pH Sensor was calibrated using three buffer solutions with pH levels of 4.0, 6.86, and 9.18 [10]. The DO value is one of the most important factors for aquatic water quality. Therefore, an optical DO (ODO) sensor (FOPTOD Optical Dissolved Oxygen) with an accuracy of ±0.1 mg/L (@ −40 °C to 100 °C) [22] was used to calibrate the DO sensor. The Analog EC Sensor was calibrated using two conductivity solutions of 1413 μS/cm and 12.88 mS/cm [13]. The accuracy and confidence of the estimated salinity level was evaluated as compared to the measurement result by an in situ, commercially available salinity sensor [23]. 

The observation of the sensors with pH/EC buffer solutions and instruments at a 1 min sampling period for 24 h measurement is depicted in Figure 7. Taking the pH/EC buffer solutions and the measurement results of the calibration instruments as references, the relative difference between the measurement and calibration results is defined as
(10)$$e_{i}^{j}=x_{i}^{j}-{\overset{¯}{x}}_{i}^{j}$$
where $x_{i}^{j}$ and ${\overset{¯}{x}}_{i}^{j}$ are the *j*^th^ measurement values of the multi-sensor module and measurement instruments and *i* = *T* (temperature); pH, DO, EC, and salinity. The performance indexes of both the mean absolute error (MAE) and root mean square error (RMSE) for accuracy analysis are defined as
(11)$${\mathrm{MAE}}_{i}=\frac{1}{N}\overset{N}{\underset{j=1}{\sum }}\left|x_{i}^{j}-{\overset{¯}{x}}_{i}^{j}\right|$$
and
(12)$${\mathrm{RMSE}}_{i}=\sqrt{\frac{1}{N}\overset{N}{\underset{j=1}{\sum }}{\left(x_{i}^{j}-{\overset{¯}{x}}_{i}^{j}\right)}^{2}}$$
where *N* is the observation number. As shown in Figure 7a, the reading difference between the DS18B20 temperature sensor and HEL-711 RTD ranged from −0.10 °C to 0.10 °C, and the corresponding MAE and RMSE values were 0.0331 °C and 0.0403 °C, respectively. This means that the DS18B20 waterproof temperature sensor, compared to the RTD sensor, has enough confidence in the operation environment of freshwater aquaculture with an accuracy of ≤±0.5 °C. Taking the pH buffer solutions of 4.0, 6.86, and 9.18 as references, the difference comparisons for the analog pH sensor were in the ranges of −0.26–0.31, −0.20–0.16, and −0.31–0.23, respectively. Furthermore, the corresponding MAE and RMSE (MAE/RMSE) values were 0.0889/0.1094, 0.0624/0.0755, and 0.0920/0.1119, respectively. From the above accuracy analysis with pH buffer solutions, the analog pH sensor has better performance for freshwater aquaculture, which is caused by the offset voltage being near the pH level of 7.0. As compared with the FOPTOD ODO, the difference range for the galvanic DO sensor ranged from −0.10 mg/L to 0.10 mg/L; moreover, the corresponding MAE and RMSE values were 0.0339 mg/L and 0.0414 mg/L, respectively. The better performance of the galvanic DO sensor was attributed to the static flow of water. Referring to both EC buffer solutions of 1413 μS/cm and 12.88 mS/cm, the differences for the analog EC sensor were in the range of −0.19–0.17 mS/cm and −0.28–0.26 mS/cm, respectively. In addition, the corresponding MAE and RMSE (MAE/RMSE) values were 0.0574/0.0698 mS/cm and 0.0882/0.1082 mS/cm, respectively. With reference to the measurement results of the EC sensor and EC buffer solution of 1413 μS/cm, the estimated salinity results were compared to the measurement ones by a Vernier Salinity Sensor. The salinity differences between the estimated and measured results ranged from −10 ppm to 9 ppm, and the corresponding MAE and RMSE values were 3.0285 ppm and 3.6451 ppm, respectively. It should be noted that the salinity level can be estimated in a cost-effective manner with sufficient accuracy by the EC sensor for freshwater aquaculture based on Equation (9). The accuracy analysis results between the proposed multi-sensor module and measurement instruments in the pre-calibration process are tabulated in Table 5. The results show that these relative differences are all small enough to be acceptable as compared to the accuracy of the measurement instruments. These illustrate the reading accuracy and measurement confidence of the proposed multi-sensor system.

### 4.2. In Situ Monitoring Results

In order to evaluate the performance of the proposed wireless multi-sensor water quality monitoring system for aquaculture, a consecutive 20-day in situ evaluation was conducted in the H715 Lab. Room, Da-Yen University in Taiwan on 1 August to 20 August 2021. All on-site observations of water temperature, pH, DO, and EC were acquired at a 10 min interval. The measured data and the estimated salinity level were sent to a local Wi-Fi AP by the ESP-WROOM-32 Wi-Fi module and were further transmitted to the ThingSpeak IoT platform (Channel ID:1408273, available online: https://thingspeak.com/channels/1408273, accessed on 27 November 2021). Figure 8a shows a small variation in water temperature measurement at about 25.1 °C. Taking the average of water temperature as a reference, the variation range of the water temperature was in the range of −3.1 °C to −3.4 °C, which could be caused by environmental disturbances the temperature sensor worsened The extension of the min–max difference between the measured and mean temperature caused the corresponding MAE and RMSE values to increase (MAE*_T_* = 0.0961 °C, RMSE*_T_* = 0.0961 °C). The pH level as shown in Figure 8b slightly increases as the measurement process progresses, which could be caused by the natural evaporation of water. Therefore, the circulation and supplement of water for aquaculture is of importance to maintain the aquatic pH level. The difference between the measured and mean pH level ranged from −0.20 to 0.20, and the corresponding MAE and RMSE values were 0.0640 and 0.0782, respectively. Furthermore, the measurement results in Figure 8c illustrate that the water quality naturally declines, with the DO level descending without external aeration. Figure 8d presents the low EC level of freshwater and the satisfactory performance of the analog EC sensor. The average EC level of the freshwater was 0.7239 mS/cm, the difference between the measured and mean EC level and was in the range of −0.18 ms/cm to 0.18 ms/cm, and the corresponding MAE and RMSE values were 0.0571 ms/cm and 0.0694 ms/cm, respectively. The EC has a sufficient accuracy for freshwater aquaculture; 

Figure 8e demonstrates an accurate and confident estimation for the aquatic salinity. The average salinity level of the freshwater was 464 ppm, the difference range between the measured and mean EC level was ±2 ppm, and the corresponding MAE and RMSE values were 0.5337 ppm and 0.7557 ppm, respectively. The accuracy analysis for temperature, pH, DO, EC, and salinity level for the in-situ measurement is listed in Table 6. The in situ measurements with a sampling rate of 10 min intervals for 20 continuous days have demonstrated the accuracy, confidence, and durability of the proposed multi-sensor wireless IoT water quality monitoring system for freshwater aquaculture.

### 4.3. Post-Calibration Results

After the 20-day on-site monitoring process, the proposed wireless multi-sensor system was calibrated with the same temperature, pH, ODO, EC, and salinity sensors as illustrated in the pre-calibration process. In order to verify the reliability and confidence of the proposed multi-sensor system, the post-calibration process was carried out for freshwater aquaculture at the same environment and conditions for each sensor. The measurement results for both the multi-sensor module and measurement instruments are depicted in Figure 9. During the 20-day measurement from 1 August to 20 August 2021, only ambient temperature was controlled at 25 °C Figure 9 depicts continuous 24 h readings for 10 days from 10 August 2021 to monitor the water quality of the in situ aquaculture. As compared to those in Figure 7, there are no obvious differences existing for the water temperature, pH, DO, EC, and salinity readings. The details of sensing differences are also listed in Table 7, and the differences are highlighted in blue. Both MAE and RMSE for the B18D20 temperature sensor are slightly rising. The variation of the difference in pH level decreases by a small margin. In addition, the differences for both the EC sensor and salinity estimation have a slight increase.

### 4.4. Discussion

In order to easily assess the accuracy of the proposed wireless multi-sensor IoT system, the difference between the measurement results of the proposed multi-sensor system and commercial sensors in pre-calibration and post-calibration processes are shown in Figure 7 and Figure 9. The corresponding performance measures of MAE and RMSE were adopted to evaluate the accuracy. The MAE is a linear score, which means that all the individual differences are weighted equally in the average. On the other hand, The RMSE is a quadratic scoring rule which measures the average magnitude of the differences. Since the differences are squared before they are averaged, the RMSE gives a relatively high weight to large ones. This means the RMSE is the most useful measure when large differences are particularly undesirable.

Even having sufficient performance in the electrical characteristics, the possible contamination in the electrodes of individual sensors could be caused by buffer solutions and freshwater. The cleaning and proper care after measurement are of great importance for both the measurement accuracy and longevity of sensors. The possible variation effect caused by the buffer solutions and aquatic corrosion on the multi-sensor system was verified by the post-calibration. The results found that no possible pollution is caused by the proposed multi-sensor module after pre-calibration and in situ measurement. Self-cleaning and optimal care for versatile aquatic sensors will be designed in the future. The aging issue and maintenance of the proposed multi-sensor system could be further verified and pre-defined.

## 5. Conclusions

A wireless multi-sensor Internet of Things (IoT) system which integrates water temperature, pH, dissolved oxygen (DO), and electrical conductivity (EC) sensors with Wi-Fi wireless communication and demonstrates the above sensing data and an estimated salinity level by the EC level in the ThingSpeak IoT platform for freshwater aquaculture has been illustrated and evaluated with sufficient accuracy and confidence. The proposed multi-sensor IoT system has the good tolerance performance of sterile cultivation for freshwater aquaculture. From the viewpoint of smart aquaculture, the proposed wireless multi-sensor IoT system features (1) a much simpler set-up and maintenance and more cost-effectiveness than multiple single sensors without external sample-taking instruments and wiring; (2) sufficient accuracy and reliability with pre-calibration even for commercialized sensor devices; and (3) simultaneous on-site monitoring of multiple sensing parameters in close proximity to an aquatic cultivation field over weeks or even months and a substantially cost-effective improvement in labor cost. Thanks to the promising development of precision agriculture, the proposed IoT platform will provide a flexible and expandable biotechnology development to simultaneously monitor a wide range of cultivation parameters in agricultural with transparency and quality control through the whole process. The issues of ageing and maintenance requirements of aquatic sensors will be further considered for measurement robustness and user-friendliness.

## Figures and Tables

**Figure 1 sensors-21-08179-f001:**
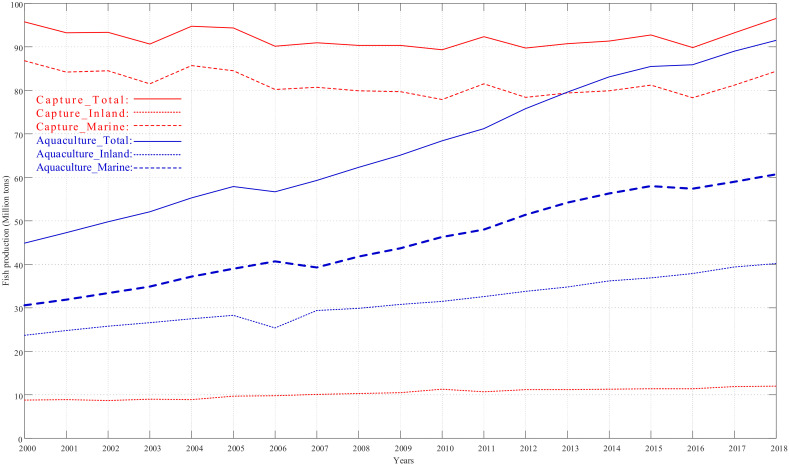
Global fish production of capture fisheries and aquaculture.

**Figure 2 sensors-21-08179-f002:**
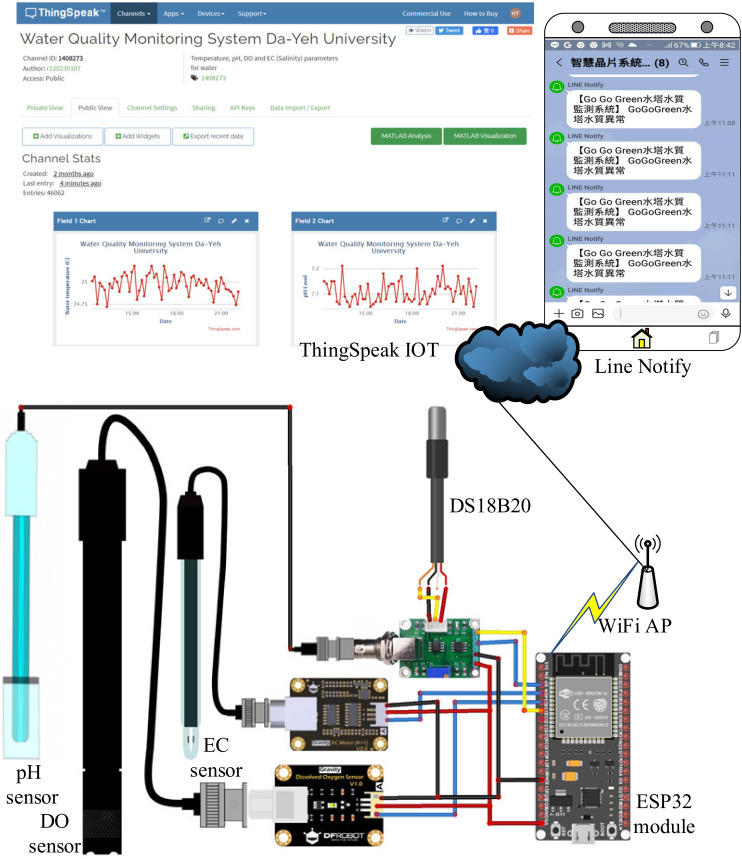
System schematic of wireless multi-sensor water quality monitoring system.

**Figure 3 sensors-21-08179-f003:**
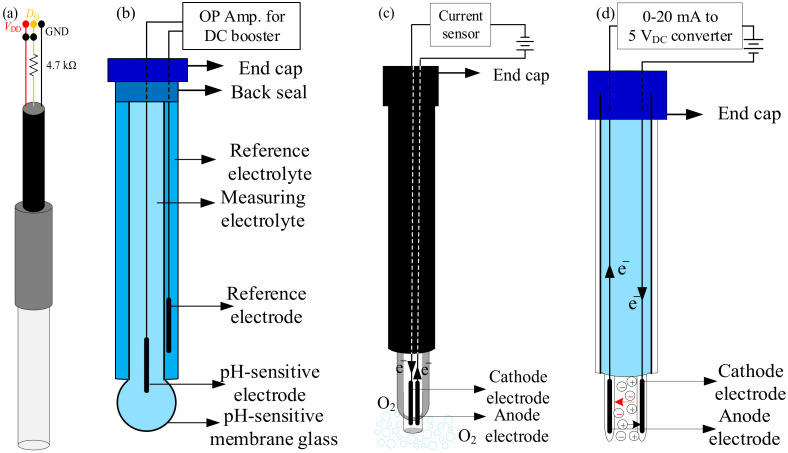
Schematics of sensors: (**a**) DS18B20 temperature sensor; (**b**) pH sensor; (**c**) DO sensor; and (**d**) EC sensor.

**Figure 4 sensors-21-08179-f004:**
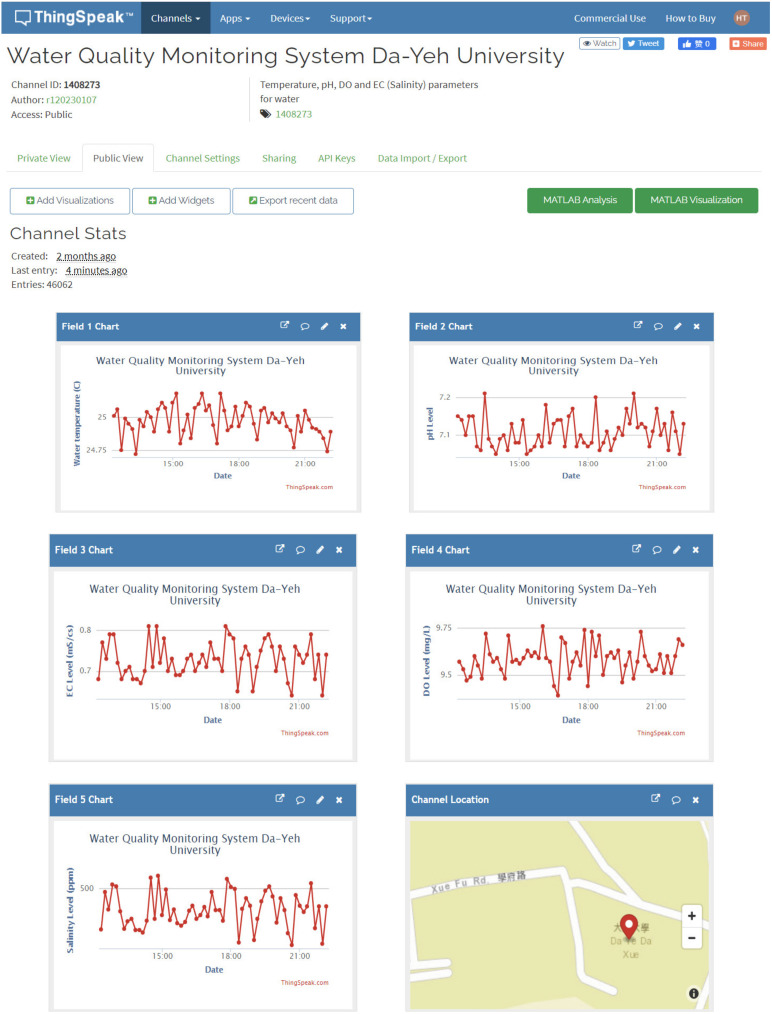
Snapshot of ThingSpeak IoT for water quality monitoring system.

**Figure 5 sensors-21-08179-f005:**
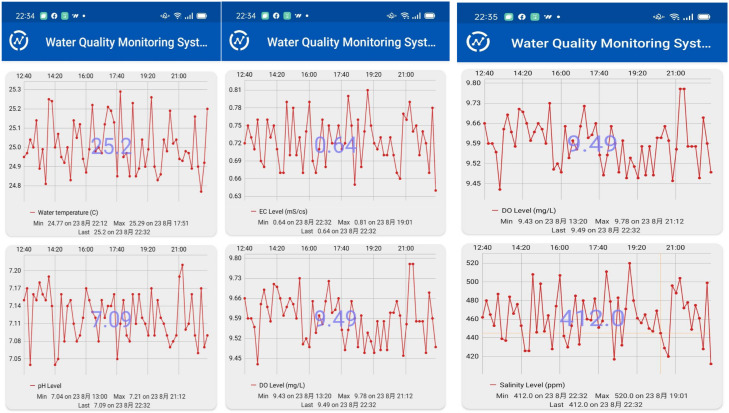
Snapshot of ThingView APP for water quality monitoring system.

**Figure 6 sensors-21-08179-f006:**
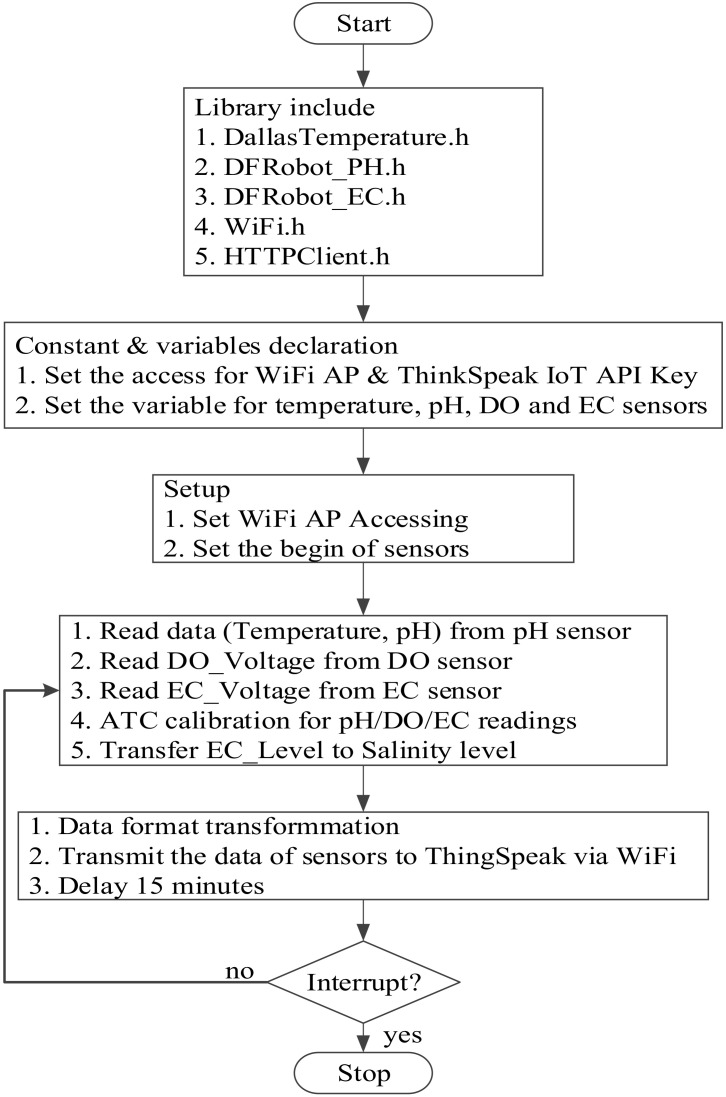
Flowchart of wireless multi-sensor module.

**Figure 7 sensors-21-08179-f007:**
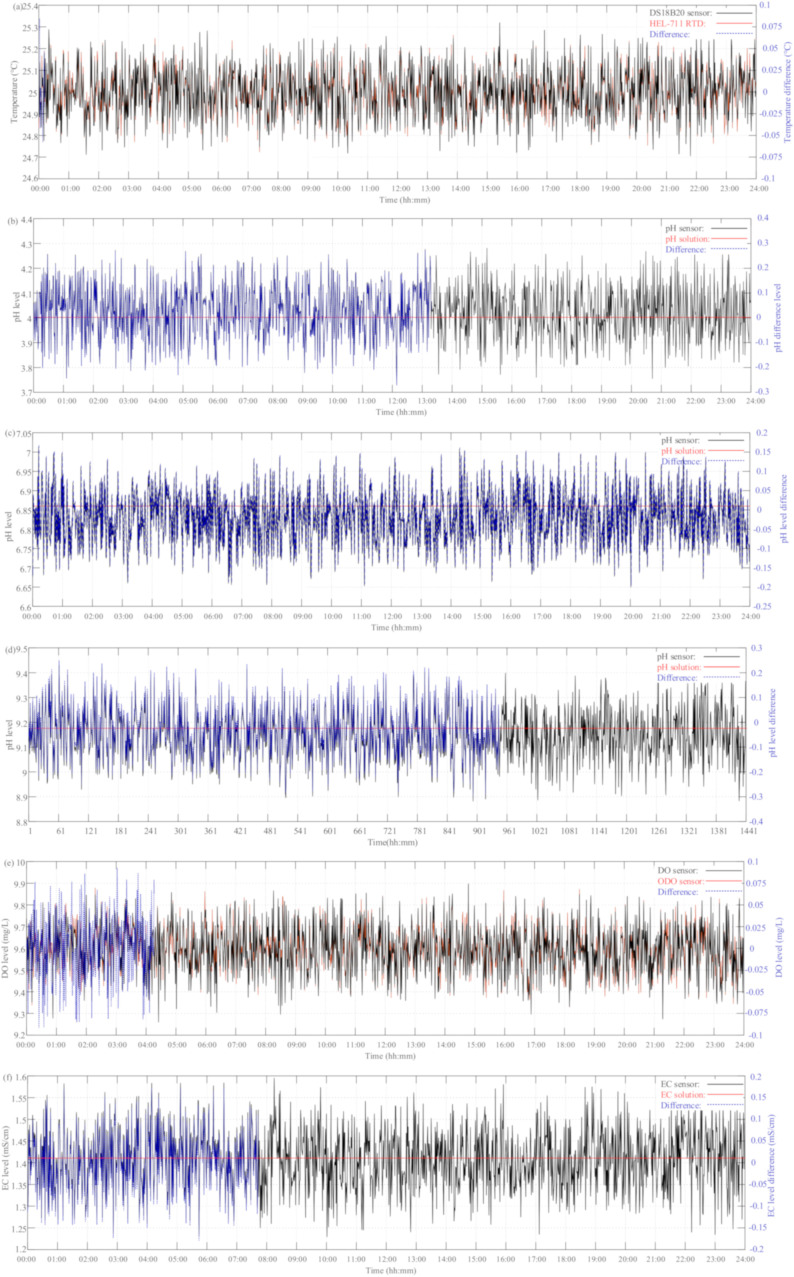
The 24 h measurement results of pre-calibration process: (**a**) DS18B20 temperature sensor vs. HEL-711 RTD; (**b**) pH sensor vs. pH 4.4 buffer solution; (**c**) pH sensor vs. pH 6.86 buffer solution; (**d**) pH sensor vs. pH 9.0 buffer solution; (**e**) DO sensor vs. FOPTOD ODO; (**f**) EC sensor vs. EC 1413 μS/cm solution; (**g**) EC sensor vs. EC 12.88 mS/cm solution; and (**h**) salinity estimation vs. Vernier salinity sensor.

**Figure 8 sensors-21-08179-f008:**
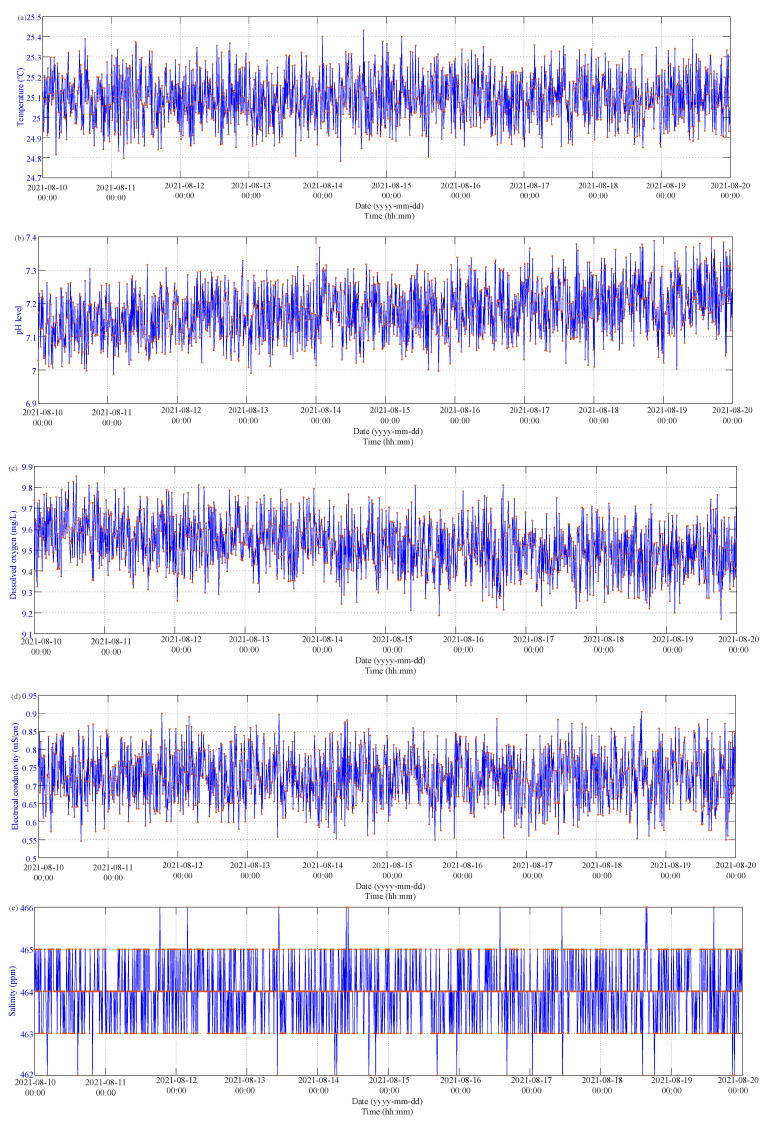
The 10-day in situ measurement results: (**a**) water temperature; (**b**) pH level; (**c**) DO level; (**d**) EC level; and (**e**) salinity level.

**Figure 9 sensors-21-08179-f009:**
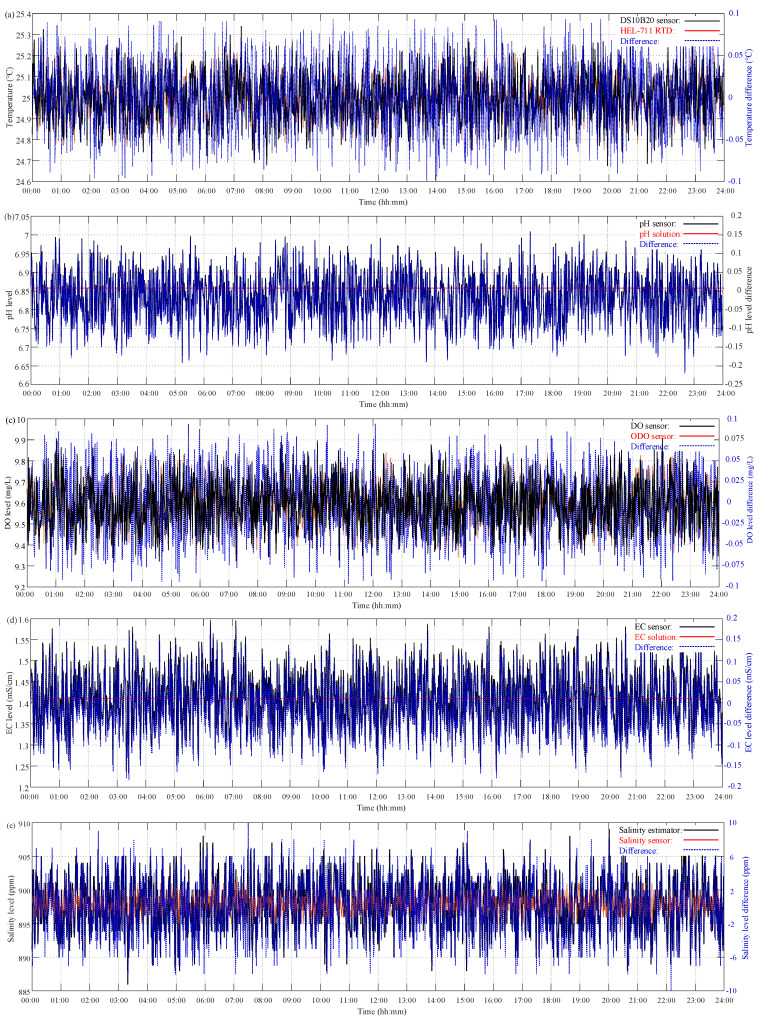
The 24 h measurement results of post-calibration process: (**a**) temperature; (**b**) pH level; (**c**) DO level; (**d**) EC level; and (**e**) salinity level.

**Table 1 sensors-21-08179-t001:** Comparisons of water quality monitoring systems for aquaculture.

No.	Sensors							Processor	Communication	Information Platform	References
	T	pH	DO	EC	TD	ORP	Sal				
1	2	1	1	1	0	0	*	PICNIC 2.0	CDMA	Intranet Platform	Zhua et al. (2010)
2	1	1	1	1	0	0	0	Multimeter	×	×	Zhang et al. (2011)
3	1	1	1	1	0	0	0	Wastmode	ZeeBee/WiFi/GPRS	DiGi International	Odey & Li (2013)
4	1	1	1	0	0	0	0	ATMega	ZeeBee	LabVIEW	Simbeye & Yang (2018)
5	3	0	1	0	0	0	0	Dataloger	Manual offload	HOBO	Schmidt et al. (2019)
6	1	1	0	1	0	1	1	Raspberry Pi	WiFi/CDMA	ThingSpeak	Saparudin et al. (2019)
7	1	1	1	0	0	1	0	Embeded MCU	RoLa/WiFi	ThingSpeak	Dahn et al. (2020)
8	1	1	1	1	0	0	1	ESP-32	WiFi	ThingSpeak	Proposed

**Table 2 sensors-21-08179-t002:** Look-up table for temperature-related modification of pH buffer solutions.

pH	pH Buffer Solutions @25 °C						
°C	1.68	4.01	6.86	7.00	9.18	10.10	12.46
0	1.67	4.01	6.98	7.12	9.46	10.32	13.47
5	1.67	4.01	6.95	7.09	9.39	10.25	13.25
10	1.67	4.00	6.92	7.06	9.32	10.18	13.03
15	1.67	4.00	6.90	7.04	9.27	10.12	12.83
20	1.68	4.00	6.88	7.02	9.22	10.06	12.64
25	1.68	4.01	6.86	7.00	9.18	10.01	12.46
30	1.69	4.01	6.85	6.98	9.14	9.97	12.29
35	1.69	4.02	6.84	6.98	9.10	9.93	12.14
40	1.70	4.03	6.84	6.97	9.07	9.89	11.99
45	1.70	4.04	6.83	6.97	9.04	9.86	11.86
50	1.71	4.06	6.83	6.97	9.01	9.83	11.73
55	1.72	4.08	6.83	6.97	8.99	9.81	11.61

**Table 3 sensors-21-08179-t003:** Main specification and warning range of aquatic sensors.

Sensors	Measuring Range	Measuring Accuracy	Operating Temperature	Warning Range
Temperature	−55–125 °C	±0.5 °C @ −10–85 °C	−55–125 °C	≤0 °C or ≥40
pH	0–14	±0.1 @ 25 °C	0–60 °C	<6.5 or >8.5
DO	0–20 mg/L	NA	0–50 °C	≤6 mg/L
EC	0–20 mS/cm	±5% F.S.	0–40 °C	≥5 mS/cm

**Table 4 sensors-21-08179-t004:** Pin assignment of ESP32 Wi-Fi module for proposed multi-sensor system.

Pin No.	Pin Function	Function	Remark
3	GPIO36 (ADC0)	pH reading	Input only
4	GPIO39 (ADC3)	DO reading	Input only
5	GPIO34 (ADC6)	EC reading	Input only
6	GPIO35 (ADC7)	Temperature reading	Input only
14	GND	Ground	
19	5 V_DC_	Power supply	

**Table 5 sensors-21-08179-t005:** Accuracy analysis of water temperature, pH, DO, and EC, and salinity levels for pre-calibration.

Items	Reference	Accuracy Analysis		
		Difference Range	MAE	RMSE
Temperature sensor	HEL-711 RTD	−0.10–0.10 °C	0.0331 °C	0.0403 °C
pH sensor	pH 4.0 buffer solution	−0.26–0.31	0.0899	0.1094
	pH 6.86 buffer solution	−0.20–0.16	0.0624	0.0755
	pH 9.18 buffer solution	−0.31–0.23	0.0920	0.1119
DO sensor	FOPTOD ODO	−0.10–0.10 mg/L	0.0339 mg/L	0.414 mg/L
EC sensor	1413 μS/cm conductivity solution	−0.19–0.17 mS/cm	0.0574 mS/cm	0.0698 mS/cm
	12.88 mS/cm conductivity solution	−0.28–0.26 mS/cm	0.0882 mS/cm	0.1082 mS/cm
Salinity estimation	Vernier Salinity Sensor	−10–9 ppm	3.0285 ppm	3.6451 ppm

**Table 6 sensors-21-08179-t006:** Accuracy analysis of water temperature, pH, DO, and EC, and salinity levels for in situ measurement.

Items	Accuracy Analysis			
	Mean	Difference Range	MAE	RMSE
Temperature sensor	25.0951 °C	−0.31–0.34 °C	0.0961 °C	0.1178 °C
pH sensor	7.1755	−0.20–0.20	0.0640	0.782
DO sensor	9.5228 mg/L	−0.35–0.33 mg/L	0.0990 mg/L	0.1217 mg/L
EC sensor	0.7923 mS/cm	−0.18–0.18 mS/cm	0.0571 mS/cm	0.0694 mS/cm
Salinity estimation	464 ppm	−2–2 ppm	0.5337 ppm	0.7557 ppm

**Table 7 sensors-21-08179-t007:** Accuracy analysis of water temperature, pH, DO, and EC, and salinity levels for post-calibration.

Items	Reference	Accuracy Analysis		
		Difference Range	MAE	RMSE
Temperature sensor	HEL-711 RTD	−0.10–0.10 °C	0.0339 °C	0.0417 °C
pH sensor	pH 6.86 buffer solution	−0.23–0.15	0.0603	0.0738
DO sensor	FOPTOD ODO	−0.10–0.10 mg/L	0.0329 mg/L	0.401 mg/L
EC sensor	1413 μS/cm conductivity solution	−0.19–0.17 mS/cm	0.0574 mS/cm	0.0698 mS/cm
Salinity estimation	Vernier Salinity Sensor	−10–10 ppm	2.9743 ppm	3.6078 ppm

## Data Availability

Not applicable.

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
