# Peer review of "An Integrated Wireless Multi-Sensor System for Monitoring the Water Quality of Aquaculture"

_sensors, 2021, doi:10.3390/s21248179_

Round 1
Reviewer 1 Report
This paper proposes a wireless multi-sensor system by integrating the temperature, pH, DO, and EC sensors with an ESP 32 WiFi module for monitoring water quality of freshwater aquaculture, which acquire the sensing data and salinity information directly derived from EC level.
The experimental results illustrate the effectiveness of the proposed system.
This paper has good application value. However, there are some problems in this paper:
1. Why is there only data before 2018 in Figure 1. It's 2021 now.
2. The contribution of this paper should be summarized in Section 1.
3. For the explanation after the equation, there should be no space before "where".
4. In Section 3, the content of system implementation is too little and needs to be expanded.
5. Figure 5 is not clear.
6. Each symbol in the equation should be explained. What does i mean in Eq. (11) and (12)?
7. https://thingspeak.com/channels/1408273. This web page has problems and cannot be displayed completely.
8. A thorough proofread is needed before submission. There are some formatting errors in the paper.
Author Response
Responses to Reviewer #1:
- Why is there only data before 2018 in Figure 1. It's 2021 now.
Ans: According to the open report: The State of World Fisheries and Aquaculture 2020, published by the Food and Agriculture Organization of the United Nations (FAO), available online: https://doi.org/10.4060/ca9229en, only data before 2018 (included) in page 3 are available. We have made our best effort to double-check official data online and no new one can be found.
- The contribution of this paper should be summarized in Section 1.
Ans: The contribution of our paper was summarized and highlighted in bold font in Section 1. Many thanks for kind remind.
- For the explanation after the equation, there should be no space before "where".
Ans: The errors were modified and highlighted in bold font. Thank you for kind remind.
- In Section 3, the content of system implementation is too little and needs to be expanded.
Ans: Following the kind suggestion, we have increase the content of hardware and software implementation in Section 3, which were highlighted in bolt font.
- Figure 5 is not clear.
Ans: The readability of Figure 5 was improved to our best.
- Each symbol in the equation should be explained. What does i mean in Eq. (11) and (12)?
Ans: The symbol of “i" was defined to stand for water temperature, pH, DO, EC and salinity levels, which has been explained after Eq. (10).
- https://thingspeak.com/channels/1408273. This web page has problems and cannot be displayed completely.
Ans: The web page for the Water Quality Monitoring System in the ThingSpeak IOT Platform was predefined in public, which can be searched by tag “1408273”. The readers interested can easily accessed to the web page as shown in the following shopsnap.
- A thorough proofread is needed before submission. There are some formatting errors in the paper.
Ans: We have followed the reviewer’s suggestions to double-check the correctness for our paper. The revised parts are all highlight in bold font.
I am sincerely thankful for your valuable suggestions and comments. Send my best wishes to you for the upcoming Christmas and New Year holidays.
Best Regards
Huan-Liang Tsai
Professor
Department of Computer Science and Information Engineering,
Da-Yeh University in Taiwan

Reviewer 2 Report
The authors should address the hypothesis more clearly. Currently, the manuscript is more like a report instead of a research article.
There is too little comparison of the results with other relevant studies. Water quality sensors have been applied in the field of natural water monitoring for many years. They should be enough results to be compared with, especially on the precision and accuracy of the sensors.
There are many inconsistencies of character fonts and size in the manuscript, e.g., in Section 2.1. The authors should revise them carefully.
What are the differences between pre-calibration and post-calibration processes? Are they only implemented before and after the continuous measurements? The authors mentioned using post-calibration to check the possible drift in instrument accuracy. Is there any drift?
Are there any other measurements from other methods for validation of the consecutive measurements? The authors claimed that pH and DO drifts at the end of the monitoring period represented the reality. But there is no direct evidence for such a declaim.
There are multiple typing errors. The authors should go through the manuscript carefully.
Author Response
Response to Reviewer #2:
- The authors should address the hypothesis more clearly. Currently, the manuscript is more like a report instead of a research article.
Ans: To our best knowledge from the published reference, every parameter of water quality could be visualized by commercially available sensors. In fact, the salinity level can be estimated using electrical conductivity (EC) measurement proposed in the works of Zhua, et al. (2010). However, the estimation process and results correctness were not explained in detail in their work. On the other hand, the relationship between EC and salinity in Eq. (9) proposed by the United States Salinity Laboratory Staff (1954) [15] was used to evaluate the accuracy of salinity estimation in this paper. The estimation results of salinity level were evaluated by a commercially available Salinity Sensor. The hypothesis for salinity estimation for the proposed water quality monitoring system was double-check through the process of pre-calibration, in situ measurement, and post-calibration to illustrate the accuracy and confidence.
- There is too little comparison of the results with other relevant studies. Water quality sensors have been applied in the field of natural water monitoring for many years. They should be enough results to be compared with, especially on the precision and accuracy of the sensors.
Ans: The capital cost and maintenance cost of integrated multi-sensor system for monitoring water quality are high with the increase in the precision and accuracy of sensors. This work was our debut in the field of sensing water quality and the precision and confidence were evaluated with pH/EC standard solutions and commercially available temperature, optical dissolved oxygen (ODO) and salinity sensors with high accuracy. The cost-effective and commercially available sensors for monitoring water quality with replaceable electrodes will be adopted in our future research.
- There are many inconsistencies of character fonts and size in the manuscript, e.g., in Section 2.1. The authors should revise them carefully.
Ans: Both font type and size were sincerely revised and double-checked in the revised manuscript, especially for symbols and units. The revised parts were highlight in bold font. Many thanks for the kind remind.
- What are the differences between pre-calibration and post-calibration processes? Are they only implemented before and after the continuous measurements? The authors mentioned using post-calibration to check the possible drift in instrument accuracy. Is there any drift?
Ans: As mentioned in Section 4, the pre-calibration was conducted for pH/EC sensors with pH/EC standard solution, as well as temperature and ODO sensors with high-accuracy ones to make sure the confidence of the following in situ continuous measurement. After 20-day continuous measurement in field, the all sensors were double-checked the possible drift in accuracy with the same standard solutions and calibrated sensors. We found no increasing variation in accuracy for the proposed water quality monitoring system.
- Are there any other measurements from other methods for validation of the consecutive measurements? The authors claimed that pH and DO drifts at the end of the monitoring period represented the reality. But there is no direct evidence for such a declaim.
Ans: The international standards of both ISO/IEC 17025 and ISO 15189 well regulate the laboratory requirements and propose a metrological approach to the measurement processes, requiring method validation, establishment of metrological traceability, estimation of measurement uncertainty, monitoring of trends in measurement processes etc. The bias and uncertainty issues of measurement are checked using pre-calibration. Then the repeatability and reproductability for measurement are double-checked using post-calibration. Both mean absolute error (MAE) and root mean square error (RMSE) for accuracy and precision analysis. To our best knowledge, the accuracy drift for temperature, pH DO, and EC sensors are only validated through in situ experiment using standard solutions and commercially available sensors with higher accuracy.
The other typing errors were sincerely revised and carefully double-checked. The revised parts were highlighted in bold font. I heartily appreciate your valuable suggestions and comments. Send my best wishes to you for the upcoming Christmas and New Year holidays.
Yours sincerely
Huan-Liang Tsai
Professor
Department of Computer Science and Information Engineering,
Da-Yeh University in Taiwan

Round 2
Reviewer 1 Report
The manuscript has addressed all the comments in previous review.